# Effect of Nitrogen Ion Implantation on the Tool Life Used in Particleboard CNC Drilling

**DOI:** 10.3390/ma15103420

**Published:** 2022-05-10

**Authors:** Jacek Wilkowski, Albina Jegorowa, Marek Barlak, Zbigniew Werner, Jerzy Zagórski, Bogdan Staszkiewicz, Jarosław Kurek, Michał Kruk

**Affiliations:** 1Department of Mechanical Processing of Wood, Institute of Wood Sciences and Furniture, Warsaw University of Life Sciences, 159 Nowoursynowska St., 02-776 Warsaw, Poland; jacek_wilkowski@sggw.edu.pl; 2Plasma/Ion Beam Technology Division, Material Physics Department, National Centre for Nuclear Research Świerk, 7 Sołtana St., 05-400 Otwock, Poland; marek.barlak@ncbj.gov.pl (M.B.); zbigniew.werner@ncbj.gov.pl (Z.W.); jerzy.zagorski@ncbj.gov.pl (J.Z.); bogdan.staszkiewicz@ncbj.gov.pl (B.S.); 3Department of Artificial Intelligence, Institute of Information Technology, Warsaw University of Life Sciences, 159 Nowoursynowska St., 02-776 Warsaw, Poland; jaroslaw_kurek@sggw.edu.pl (J.K.); michal_kruk@sggw.edu.pl (M.K.)

**Keywords:** high-speed steel (HSS), nitrogen ion implantation, tool life, particleboard, CNC drilling

## Abstract

The paper presents the effect of nitrogen ion implantation on the tool life of the tools commonly used in the furniture industry for drilling particleboards. Nitrogen ions with different accelerating voltages of 25, 40, 55, and 70 kV and a fluence of 5 × 10^17^ cm^−2^ were implanted into the surface of commercially available high-speed steel (HSS) drills, using the implanters without mass-separated ion beams. The tests were carried out in a computerized numerical control (CNC) machining center used in the furniture industry. Based on the measurements of the direct tool wear indicator (W), the drill wear curves were determined and the relative tool life index, standard deviation, coefficient of variation, and the implantation quality index of tool life were calculated. The studies have shown that the modification of the drill surface layer by the nitrogen ion implantation process increases the tool life. The obtained results allow the research to be the continued in a wider scope.

## 1. Introduction

The increasing use of wood-based composites in the furniture and construction industries has created a need for high-performance and complex tooling systems that are capable of resisting the aggressive wear environment associated with the machining of these materials.

Regardless of the introduction of new tool materials in wood machining, such as WC-Co composites and polycrystalline diamonds, high-speed tool steel (HSS) is still a tool material of choice in CNC drilling [1]. The popularity of this material can be explained by the lower cost, the reduction in breakage in the older spindles and hand routers, the sharper edges allowing for easier hand feeding, and the different geometries [1,2,3].

The three-layer particleboard used in the tests can be described as a material with low machinability due to an increased tool wear caused by the presence of minerals, adhesives, and fillers [4]. At the same time, this wood-based board is the most frequently used construction material in the serial, industrial production of furniture on CNC machine tools.

The wear of the cutting tools is a limiting factor in the high-performance machining of wood and wood-based panels. Tool wear affects not only tool life, but also energy consumption, surface finish, and production rate. To increase productivity and lower operating costs, advanced cutting tools are required that can provide longer tool life, less unplanned downtime, and better surface quality.

Improving the durability of cutting tools has been of interest to many researchers for many years [5,6,7]. Tools are often surface-layer treated. There are three ways to extend the tool life through such modification. These are the modification of the surface region, the deposition of an additional layer, and duplex treatment, i.e., modification of the surface region and the deposition of an additional layer [8].

In the case of steel tools used in the woodworking industry, the modification and the improvement of the properties of the surface layer were realized by one of the most known and widespread technologies of ion nitriding [9,10] and plasma immersion ion implantation [11].

The deposition of additional layers usually means the application of hard coatings produced by physical vapor deposition or chemical vapor deposition [12,13]. For the deposition of abrasion-resistant coatings, electroplating methods, vacuum arc discharge, magnetron sputtering, or their combinations were used [14]. Duplex surface treatment involves first nitriding the tool and then applying a coating to increase its durability [10]. A hard coating can be applied by means of arc evaporation or glow discharge (magnetron sputtering) [8,12]. The substrate pre-prepared by nitriding yields better adhesive properties of the hard coatings deposited on it [15,16]. This method of increasing the adhesion of the coating is widely used [17,18].

Ion implantation is a method of modifying the subsurface area [19] with a well-known mechanism for reducing wear and extending the tool’s lifetime [20,21,22]. The modified area is not a layer; therefore, there is no problem of adhesion, dimensional change, or delamination of the layer during processing [22,23,24,25,26]. Ion implantation can be a good alternative to modifying cutting tools with coatings. Additionally, the ion implantation processes can be provided at room temperature and the implanted element (samples/tools) temperature can be below 100 °C. The heating of tools to a temperature of several hundred degrees during the layer deposition processes using the “thermal” techniques may result in the loss of their properties. The ion implantation is a frequently used method of modifying metals [27] and composites [26,28,29], including ceramics [30,31].

Research on the durability of carbide blades used for milling a three-layer chipboard, subjected to implantation with nitrogen ions, showed an increase in the durability of the tools [32].

However, there are few scientific reports that concern the implantation of the drill bits used for cutting in the wood industry with ions [1]. Wilkowski examined the service life of HSS drills when drilling laminated chipboard with nitrogen ion implantation at various doses, while one type of accelerating voltage was used—60 kV. Actually, the research demonstrated an increase in tool life, but also the need for deeper research.

This work is a continuation of the above-mentioned work and presents the results of research aimed at extending the service life of the high-speed steel drills used in the processing of wood and wood-based materials by applying nitrogen ion implantation at various ion energies.

## 2. Materials and Methods

Double-edged HSS Leitz drills (Leitz GmbH and Co. KG, Stuttgart, Germany) for blind drilling, with a diameter of 10 mm and a total length of 68 mm (Figure 1), were used in the research. Tools of this type are commonly used in the furniture industry. All drills (six pieces per type of modification) were implanted with nitrogen ions using a non-mass separated beam (direct beam), generated by a semi-industrial implanter [1], from the side of the brad point bits, as shown in Figure 2.

Nitrogen of 99.999% purity was used as the source of the implanted ions. The implanted nitrogen was delivered as two kinds of ions, i.e., N_2_^+^ + N^+^, in the ratio ~1:1; so, there were two elementary charges per three atoms. For example, in the case of the N_2_ molecule implanted with the acceleration voltage of 70 kV, each atom carries the energy of 35 keV, according to the law of energy conservation. It means that, for example, for a total fluence of 5 × 10^17^ cm^−2^, the fluence of 1.65 × 10^17^ cm^−2^ is implanted with the energy of 70 keV and the fluence of 3.35 × 10^17^ cm^−2^ is implanted with the energy of 35 keV. The average charge state (ACS) is at the level of 0.67 [6].

The implanted fluences (doses) were 5 × 10^17^ cm^−2^ in all cases. The acceleration voltage of the implanted ions was 25, 40, 55, or 70 kV. The beam current density was at a level of 0.6 mAcm^−2^. The implanted drills were clamped onto a stainless-steel plate to avoid overheating effects; so, the estimated value of temperature of the implanted drills did not exceed 200 °C. The working pressure in the vacuum chamber was at a level from 2 to 5 × 10^−3^ Pa.

The ion implantation processes were preceded by Monte Carlo simulations of the main parameters of the depth profiles of the implanted elements (such as peak volume dopant concentration *N_max_*, projected range *R_p_*, range straggling Δ*R_p_*, kurtosis, and skewness) [6], using freeware type code SRIM-2013.00 (the Stopping and Range of Ions in Matter) [33]. The simulation was performed for 100 000 implanted ions of nitrogen to iron—the main component of steel. The angle of the ion incidence was defined as 0° (perpendicular to the implanted target). The simulations were performed for room temperature implantation. The modelling did not take into account the phenomenon of substrate sputtering by the implanted ions. The theoretical values of the sputtering yield Y were additionally calculated, using the commonly known freeware-type quick ion implantation calculator SUSPRE [34], from the energy deposited in the surface region of the material using the Sigmund formula.

In all cases, the simulations were performed for room temperature and for the total implanted fluence of 5 × 10^17^ cm^−2^. The detailed information about the values of the acceleration voltage and the energy of the implanted ions is presented in Table 1.

The selected elements of the microstructure in the transverse micro-sections, obtained using SEM and EDS methods, were presented in our previous paper [1]. No changes in the microstructure were observed due to the ion implantation, at this level of the observation.

The tool life tests were carried out on 30 drills of one type. Six repetitions of cutting were used for each type of implantation and reference drills. The cuttings were performed using the Centateq P-110 Homag CNC machining center (Homag Group AG, Schopfloch, Germany), as shown in Figure 3. A three-layer particleboard produced by Pfleiderer company (Grajewo, Poland) with a thickness of 18 mm and density 648 kg/m^3^ was machined. The characteristics of this material are presented in Table 2. The selected properties were determined according to the European standards (EN): the density by EN 323 [35], water absorption after 24 h by EN 317 [36], the MOR in bending and the MOE in bending by EN 310 [37], and the tensile strength perpendicular to the plane by EN 319 [38].

Workpieces with dimensions of 1000 × 400 × 18 mm^3^ were blind drilled to a depth of 15 mm with five tools simultaneously (tools from all tested modification variants were used), using drilling gear V12/H4 × 2Y, as shown in Figure 4.

Drilling was carried out for the cutting parameters recommended by the tool manufacturer, i.e., feed speed 2 m∙min^−1^ and a spindle speed of 6000 rpm. The value of the tool wear indicator *W* (the maximum size of the corner wear) was measured as shown in Figure 5. The average values of the *W* indicator of the drill were calculated from Equation (1),
*W* = (*W*′ + *W*″)/2 (1)
where *W*′ is the corner wear of the first edge of the drill (mm) and *W*″ is that of the second edge of the drill (mm).

The analysis of the tool wear was performed with the use of vision methods, based on images taken with a Canon EOS 40D digital camera, with a CMOS matrix resolution of 10.1 MPx, equipped with the Canon EF 100 mm f/2.8 Macro USM lens. The images were processed in the Gimp 2.10.24 open-source program for editing raster graphics.

The machining was conducted in the form of blunting cycles. The number of holes for particular cycles and the overall number of holes are given in Table 3. The tool life criterion was related to the value of the tool wear indicator W (defined above) and was assumed as 0.5 mm. At the moment when the drill achieved or exceeded a value of 0.5 mm in a given cycle, the cutting process was terminated, recognizing this tool as not suitable for further exploitation. As soon as this value was reached, the number of holes made by this tool was interpolated to make the result (up to the tool life criterion) more reliable. Thus, the tool life is the sum of the holes made in each blunting cycle until the tool wear value is 0.5 mm.

## 3. Results and Discussion

The modelled depth profiles of the nitrogen ions implanted in the iron substrate are shown in Figure 6. The results of the modelling of the ion implantation are presented in Table 4, whereas their change as the function of the acceleration voltage and the average ion energy (calculated taking into account the average charge state) is shown in Figure 7. Additionally, the trendlines, the equations, and the values of the coefficient of determination *R*^2^, determined using Microsoft Excel 2010 spreadsheet, are presented in Figure 7.

We can see that the value of the peak volume dopant concentration changes from 1.56 × 10^23^ cm^−2^ for the acceleration voltage of 25 kV to 6.88 × 10^22^ cm^−2^ for 70 kV. A more than twofold change has a polynomial dependence on voltage with the determination factor *R*-squared displayed in the chart.

The calculated values of sputtering yield are relatively low, and they change from 0.94 to 0.8 for the used range of the acceleration voltage. This change can be described by a 2nd degree polynomial trendline, with the significant determination factor at the level of 0.9995.

The values of the projected range, the range straggling, the kurtosis, and the skewness change in the discussed range display linear dependence on voltage. The values of the projected range and the range straggling increase with the growth of the acceleration voltage, and they are at a level of tens of nanometers. This is a typical trend. In contrast, the values of kurtosis and skewness decrease with the growth of the acceleration voltage. The relatively high values of the kurtosis and the skewness are caused by the non-homogeneity of the nitrogen ion beam.

Figure 8a shows the tool wear curves, i.e., a dependence of tool wear (direct tool wear indicator—*W*) on the number of holes made with each tool during the test. Thirty tools were tested. It should be emphasized that the wear curves of virgin (un-implanted) drills have a different shape as compared to the implanted drills. The primary/initial wear zone is clearly visible, with rapid initial wear, as shown in Figure 9, where the shape of the wear curve is in line with the classic wear theory (Lorenz wear curve), as described in detail in ref. [32]. The wear curves for the implanted tools do not follow this theory. The wear relationship is clearly linear for the tool life, and there is no characteristic rapid initial wear in the primary wear zone (Figure 8a).

Figure 8b shows the statistical values of the tool life. The tool life of the implanted drills was higher than that of the virgin tools independently of the acceleration voltage variant. The relative index of the tool life (*RI*) of the best modification was 1.17, which meant that the tool life was 17% higher than the virgin tools group. The best implantation variant was the one obtained for the acceleration voltage of 70 kV, but the differences between the means for the implanted tools groups were not statistically significant.

The coefficient of variation of tool life (*CV*) of the virgin drills was more than twice as high as the worst variant of the implanted drills. The coefficient of variation of tool life is of great importance in industrial practice. For drills that wear in a similar time, i.e., with a low value of the coefficient of variation, it is easier to automate the machining process than when the coefficient of variation of tool life is large. An interesting effect of implantation is a significant reduction in the coefficient of variation of tool life (Figure 8b).

Objectively, the quality of the tool modification process can be characterized by the quality index of tool life (*QI*), being the ratio of the relative index of tool life (*RI*) and the coefficient of variation of tool life (*CV*):*QI* = *RI*/*CV*(2)
where *QI*—the quality index of tool life, *RI*—the relative index of tool life, and *CV*—the coefficient of variation of tool life.

The quality index is a very important industrial parameter, next to the relative index of tool life. The higher the quality index (*QI*) value, the better the tools in terms of industrial utility. In the group of tested tools, as shown in Figure 8b, the tool modified with an acceleration voltage 40 kV is of the highest quality. The highest *QI* index was almost six times higher than the *QI* index obtained for the virgin tools group.

As mentioned earlier, the wear curves of the virgin drills have a characteristic classic shape of the tool wear, corresponding to the Lorenz wear curve (Figure 9), where there is the first zone (break-in period), with rapid initial tool wear, and the second zone (steady-state wear region), in which the increase of the direct wear indicator *W* is slow and has a linear characteristic. In contrast, the wear curves of implanted drills have a linear characteristic of tool life (Figure 8). These observations were used to model the wear curves for two variants of virgin and implanted drills (Figure 10). A linear regression model was used. With the number of drilled holes equal to 200, a clear linear characteristic of the wear curves in two analyzed groups of tools (virgin and implanted) is visible. Thus, the effect of nitrogen ion implantation was to eliminate rapid wear in the primary wear zone of the wear curve. It should be emphasized that no microscopic observations showed any wear mechanisms other than the abrasive mechanism. Other authors also drew attention to the dominance of the abrasive wear mechanism in HSS tools during the machining of wood materials [1,9,11,12,14,39,40,41].

The above results can be useful in the wood industry. Unfortunately, they will not be directly transferred, e.g., into the metal industry, due to the high differences in the structures and properties of the machined materials and the differences in the values of the machining parameters.

A part of the described issues was presented in previous papers, e.g., [7,32]. These concerned the milling operations using WC-Co indexable knives. So, it is possible to transfer the obtained results to other types of mechanical processing, but only in the wood industry.

## 4. Conclusions

Based on the results, the following conclusions can be drawn:The use of nitrogen ion implantation improved the wear resistance of the HSS drills and extended their tool life during CNC drilling of particleboard. The average tool life extension of the best implanted tool life variant was 17% as compared to that of the virgin tools.In the groups of implanted drills, the largest tool life was obtained for the modification variant with an acceleration voltage of 70 kV. The differences between the average results for the implanted groups were not statistically significant.Nitrogen ion implantation with an acceleration voltage of 40 kV is characterized by the highest quality of the modification process described by the quality index (*QI*). The index *QI* for this group of drills was 17.88, while for the virgin tools it was 3.08.The wear curves for the implanted tools do not follow the classic wear theory (the Lorenz wear curve). They are clearly linear, and no rapid initial wear in the break-in period occurs.

Based on the previous experience, the authors plan to extend the works of the implanted drills. The investigations of the ion implantation of carbide drills will be next.

## Figures and Tables

**Figure 1 materials-15-03420-f001:**
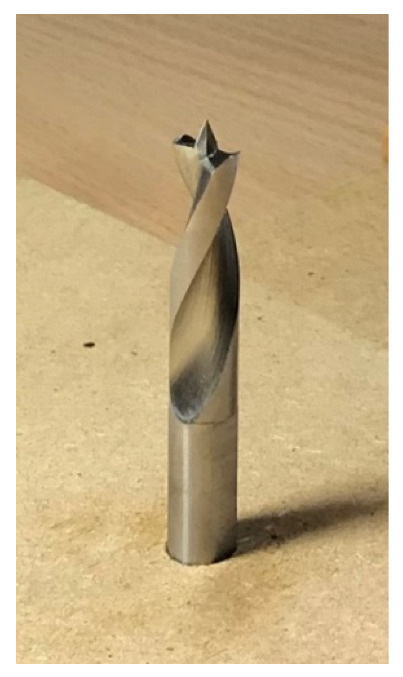
Leitz HSS drill bit.

**Figure 2 materials-15-03420-f002:**
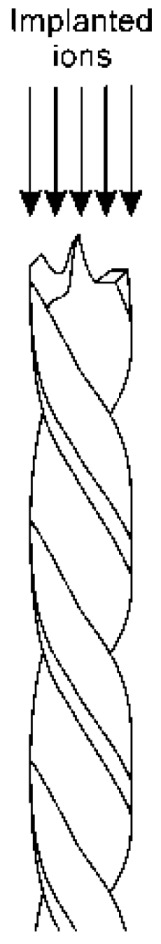
Ion beam direction during drill implantation.

**Figure 3 materials-15-03420-f003:**
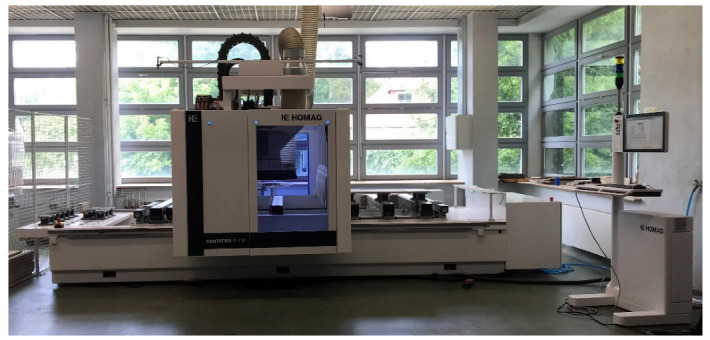
CNC machining center Homag Centateq P—110.

**Figure 4 materials-15-03420-f004:**
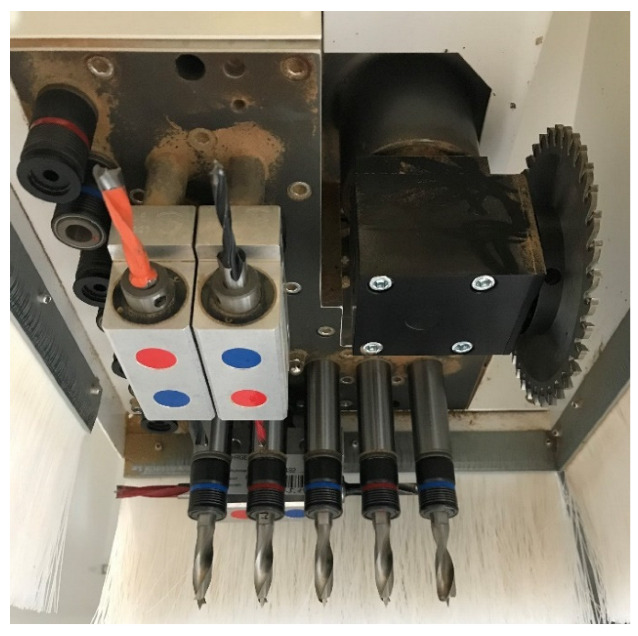
Homag drilling gear V12/H4 × 2Y.

**Figure 5 materials-15-03420-f005:**
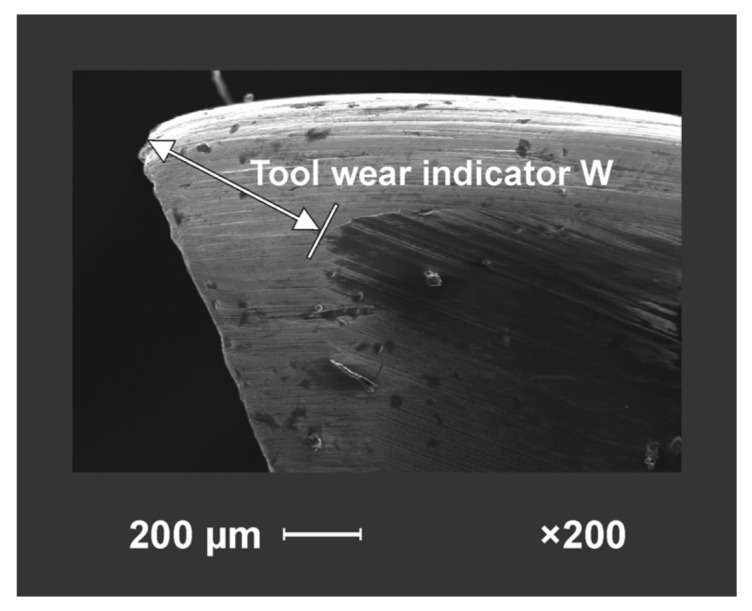
Measurement of tool wear indicator.

**Figure 6 materials-15-03420-f006:**
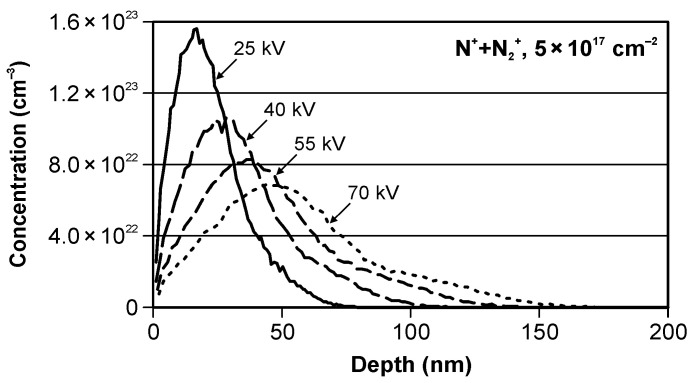
The modelled nitrogen depth profiles in iron.

**Figure 7 materials-15-03420-f007:**
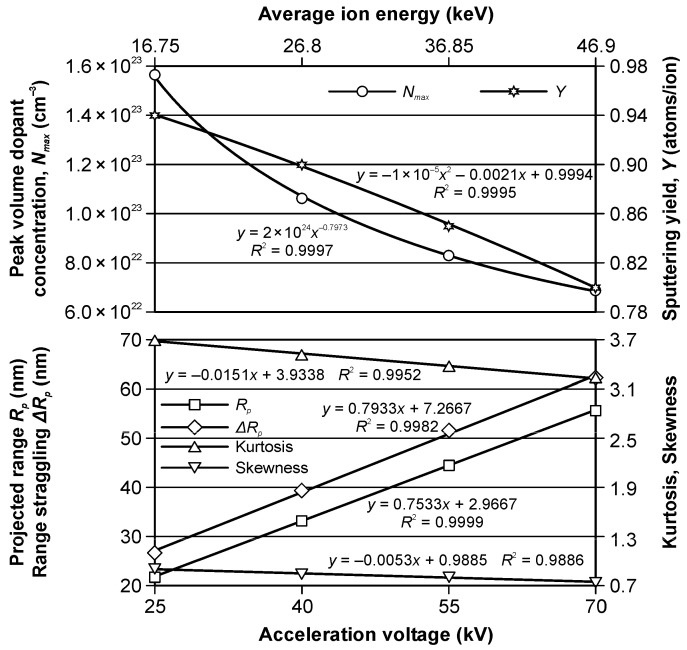
The modelled values of peak volume dopant concentration, sputtering yield, projected range, range straggling, kurtosis, and skewness as functions of the acceleration voltage and the average ion energy.

**Figure 8 materials-15-03420-f008:**
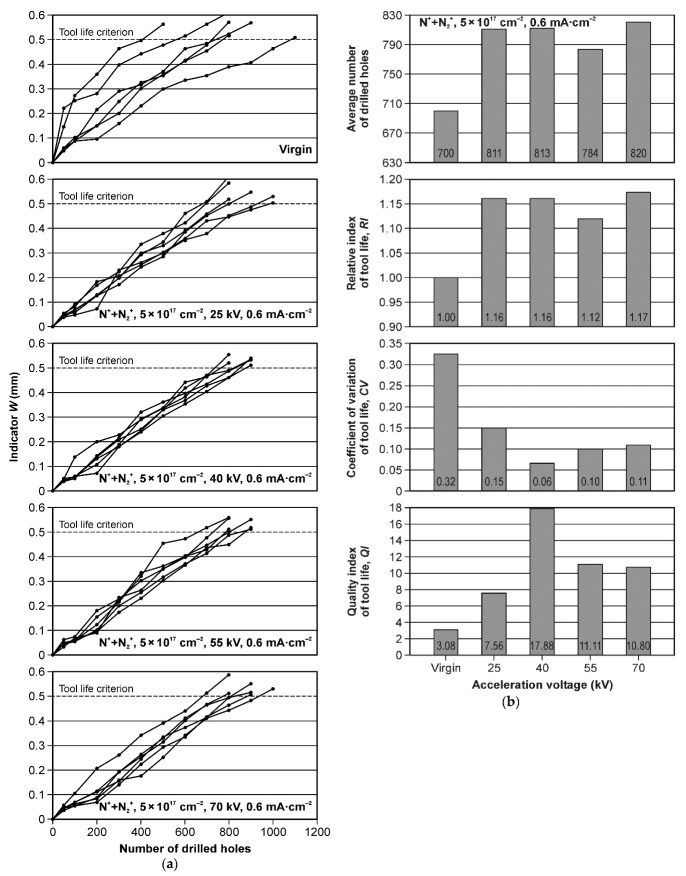
(**a**) The wear curves of the tested HSS drills; (**b**) the statistical values of tool life.

**Figure 9 materials-15-03420-f009:**
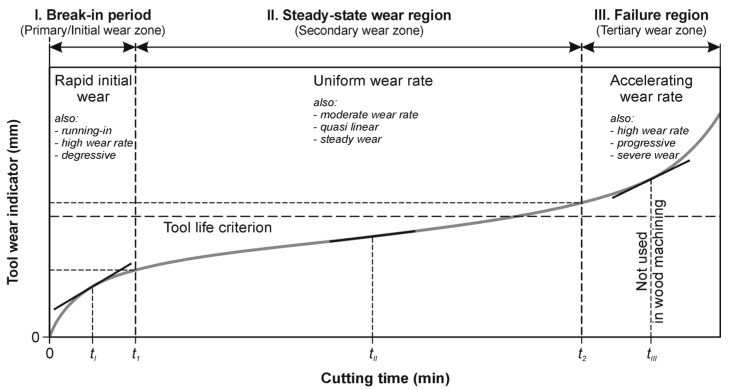
The classic wear curve (Lorenz) [33].

**Figure 10 materials-15-03420-f010:**
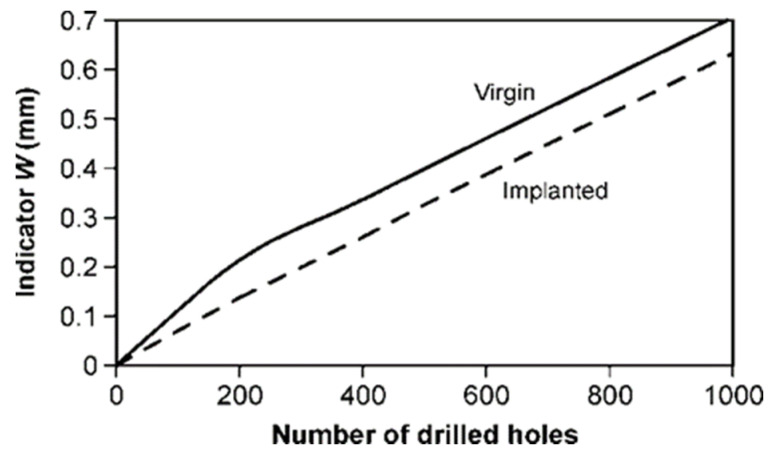
Modeled wear curves of virgin and implanted drills.

**Table 1 materials-15-03420-t001:** The information about the values of the acceleration voltage and the energy of the implanted ions.

Acceleration Voltage (kV)	Percentage Charge State Distribution (%)	
	N_2_^+^	N^+^
	67	33
	Energy (keV)	
25	12.5	25
40	20	40
55	27.5	55
70	35	70

**Table 2 materials-15-03420-t002:** Selected mechanical and physical properties of the tested three-layer particleboard.

Material	Density (kg/m^3^)	Tensile Strength (MPa)	Swelling after 24 h (%)	Flexural Strength MOR (MPa)	Modulus of Elasticity MOE (MPa)
The three-layer Particleboard	648	0.41	20.5	8.68	2212

**Table 3 materials-15-03420-t003:** Number of holes drilled in successive blunting cycles.

Cycle No.	1	2	3	4	5	6	7	8	9	10	11	12
Holes count	50	50	100	100	100	100	100	100	100	100	100	100
Total	50	100	200	300	400	500	600	700	800	900	1000	1100

**Table 4 materials-15-03420-t004:** The values of the parameters of nitrogen peak and the sputtering yield.

Acceleration Voltage (kV)	Peak Volume Dopant Concentration *N_max_* (cm^−3^)	Projected Range *R_p_* (nm)	Range Straggling Δ*R_p_* (nm)	Skewness	Kurtosis	Sputtering Yield *Y* (Atoms/Ion)
25	1.56 × 10^23^	21.7	26.6	0.8999	3.6942	0.94
40	1.06 × 10^23^	33.2	39.4	0.8435	3.5118	0.9
55	8.3 × 10^22^	44.5	51.6	0.8055	3.3769	0.85
70	6.88 × 10^22^	55.6	62.2	0.7366	3.2345	0.8

## Data Availability

Not applicable.

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
