# Peer review of "Effect of Nitrogen Ion Implantation on the Tool Life Used in Particleboard CNC Drilling"

_materials, 2022, doi:10.3390/ma15103420_

Round 1
Reviewer 1 Report
The paper presents the effect of nitrogen ion implantation on the tool life commonly used in the furniture industry for drilling particleboards. The studies have shown that the modification of drill surface layer by the nitrogen ion implantation process increases the tool life. The work offers a very interesting study. Some points need to be considered before publishing this work:
1- The problem statement needs to be improved.
2- A comparison with other available techniques should be added in the introduction section.
3- The authors should clarify how the findings of this study can be beneficial to metal cutting industry.
4- More discussion about the future work should be added.
5- Can such approached be used/implemented to the milling operations? This is very important question and needs to be fully addressed.
6- More recent papers should be added.
Reviewer 2 Report
Indeed, the method of ion implantation is well known and widely used to increase the corrosion and mechanical resistance of the surface layers of various structural materials. Therefore, to confirm the scientific novelty of the research, the authors had to show
- how the microstructure, phase state, adhesive strength and mechanical properties of the synthesized surface layers.
- how these parameters change depending on the ion implantation regimes.
To do this, it was necessary to investigate the transverse micro-sections of the working part of the cutting tool.
Besides
- 1, 2, 9 are not very informative for a scientific article and in my opinion you can do without them.
- self-citations should be reduced in the list of sources used
Round 2
Reviewer 2 Report
Unfortunately, not all of my comments have been corrected. But the reasoned answers showed that in the course of further research, the experiment will be planned more consistently.